# Provably tuning the ElasticNet across instances

**Maria-Florina Balcan**    **Mikhail Khodak**    **Dravyansh Sharma**    **Ameet Talwalkar**

Carnegie Mellon University*

## Abstract

An important unresolved challenge in the theory of regularization is to set the regularization coefficients of popular techniques like the ElasticNet with general provable guarantees. We consider the problem of tuning the regularization parameters of Ridge regression, LASSO, and the ElasticNet across multiple problem instances, a setting that encompasses both cross-validation and multi-task hyperparameter optimization. We obtain a novel structural result for the ElasticNet which characterizes the loss as a function of the tuning parameters as a piecewise-rational function with algebraic boundaries. We use this to bound the structural complexity of the regularized loss functions and show generalization guarantees for tuning the ElasticNet regression coefficients in the statistical setting. We also consider the more challenging online learning setting, where we show vanishing average expected regret relative to the optimal parameter pair. We further extend our results to tuning classification algorithms obtained by thresholding regression fits regularized by Ridge, LASSO, or ElasticNet. Our results are the first general learning-theoretic guarantees for this important class of problems that avoid strong assumptions on the data distribution. Furthermore, our guarantees hold for both validation and popular information criterion objectives.

## 1 Introduction

Ridge regression [30, 43], LASSO [41], and their generalization the ElasticNet [28] are among the most popular algorithms in machine learning and statistics, with applications to linear classification, regression, data analysis, and feature selection [15, 46, 28, 20, 24]. Given a supervised dataset $(X, y) \in \mathbb{R}^{m \times p} \times \mathbb{R}^m$ with $m$ datapoints and $p$ features, these algorithms compute the linear predictor

$$\hat{\beta}^{(X,y)}_{\lambda_1,\lambda_2} = \underset{\beta \in \mathbb{R}^p}{\arg\min} \|y - X\beta\|_2^2 + \lambda_1 \|\beta\|_1 + \lambda_2 \|\beta\|_2^2 \tag{1}$$

Here $\lambda_1, \lambda_2 \geq 0$ are *regularization coefficients* constraining the $\ell_1$ and $\ell_2$ norms, respectively, of the model $\beta$. For general $\lambda_1$ and $\lambda_2$ the above algorithm is the ElasticNet, while setting $\lambda_1 = 0$ recovers Ridge and setting $\lambda_2 = 0$ recovers LASSO.

These coefficients play a crucial role across fields: in machine learning controlling the norm of $\beta$ implies provable generalization guarantees and prevent over-fitting in practice [34], in data analysis their combined use yields parsimonious and interpretable models [28], and in Bayesian statistics they correspond to imposing specific priors on $\beta$ [35, 33]. In practice, $\lambda_2$ regularizes $\beta$ by uniformly shrinking all coefficients, while $\lambda_1$ encourages the model vector to be sparse. This means that while they do yield learning-theoretic and statistical benefits, setting them to be too high will cause models to under-fit the data. The question of how to set the regularization coefficients becomes even more unclear in the case of the ElasticNet, as one must juggle trade-offs between sparsity, feature correlation, and bias when setting both $\lambda_1$ and $\lambda_2$ simultaneously. As a result, there has been intense empirical and theoretical effort devoted to automatically tuning these parameters. Yet the

---

*Correspondence: `dravyans@cs.cmu.edu`. Author emails: `{ninamf,dravyans}@cs.cmu.edu`, `{khodak,talwalkar}@cmu.edu`

state-of-the-art is quite unsatisfactory: proposed work consists of either heuristics without formal guarantees [26, 31], approaches that optimize over a finite grid or random set instead of the full continuous domain [17], or analyses that involve very strong theoretical assumptions [44].

In this work, we study a variant on the above well-established and intensely studied formulation. The key distinction is that instead of a single dataset $(X, y)$, we consider a collection of datasets or instances of the same underlying regression problem $(X^{(i)}, y^{(i)})$ and would like to learn a pair $(\lambda_1, \lambda_2)$ that selects a model in equation (1) that has low loss on a validation dataset. This can be useful to model practical settings, for example where new supervised data is obtained several times or where the set of features may change frequently [19]. We do not require all examples across datasets to be i.i.d. draws from the same data distribution, and can capture more general data generation scenarios like cross-validation and multi-task learning [45]. Despite these advantages, we remark that our problem formulation is quite different from the standard single dataset setting. Our formulation treats the selection of regularization coefficients as *data-driven algorithm design*, which is often used to study combinatorial problems [27, 3], and has connections to meta-learning [12].

Our main contribution is a new structural result for the ElasticNet Regression problem, which implies generalization guarantees for selecting ElasticNet Regression coefficients in the multiple-instance setting. In particular, Ridge and LASSO regressions are special cases. We extend our results to obtain low regret in the online learning setting, and to tuning related linear classification algorithms. In summary, we make the following key contributions:

- We formulate the problem of tuning the ElasticNet as a question of learning $\lambda_1$ and $\lambda_2$ simultaneously across multiple problem instances, either generated statistically or coming online. Our formulation captures relevant settings like cross-validation and multi-task learning.

- We provide a novel structural result (Theorem 2.2) that characterizes the loss of the ElasticNet fit. We show that the hyperparameter space can be partitioned by polynomial curves of bounded degrees into pieces where the loss is a bivariate rational function. The result holds for both the usual ElasticNet validation objective and when it is augmented with information criteria like the AIC or BIC.

- An important consequence of our structural result is a bound on the pseudo-dimension (Definition 5) for the loss function class, which yields strong generalization bounds for tuning $\lambda_1$ and $\lambda_2$ simultaneously in the statistical learning setting (Theorem 3.2). Informally, for ElasticNet regression problems with at most $p$ parameters, for any problem distribution $\mathcal{D}$, we show that $O\left(\frac{1}{\epsilon^2}(p^2 \log \frac{1}{\epsilon} + \log \frac{1}{\delta})\right)$ problem instances (or datasets) are sufficient to learn an $\epsilon$-approximation to the best $(\lambda_1, \lambda_2)$, with probability at least $1 - \delta$.

- In the online setting, we show under very mild data assumptions—much weaker than prior work—that the problem satisfies a dispersion condition [6, 9]. As a result we can tune all parameters across a sequence of instances appearing online and obtain vanishing regret relative to the optimal parameter in hindsight over the sequence (Theorem 3.3) at the rate $\tilde{O}(1/\sqrt{T})$[2] wrt the length $T$ of the sequence.

- We also give distributional and online learning results for regularized classifiers (Theorems 4.1, 4.2).

We include a couple of remarks to emphasize the generality and significance of our results. First, in our multiple-instance formulation the different problem instances need not have the same number of examples, or even the same set of features. This allows us to handle practical scenarios where the set of features changes across datasets, and we can learn parameters that work well on average across multiple different but related regression tasks. Second, by generating problem instances iid from a fixed (training + validation) dataset, we can obtain iterations (training/validation splits) of popular cross-validation techniques (including the popular leave-one-out and Monte Carlo CV) and our result implies that $\tilde{O}(p^2/\epsilon^2)$ iterations are enough to determine an ElasticNet parameter $\hat{\lambda}$ with loss within $\epsilon$ (w.h.p.) of the optimal parameter $\lambda^*$ over the distribution induced by the cross-validation splits.

**Key challenges and insights**. A major challenge in learning the ElasticNet parameters is that the variation of the solution path as a function of $\lambda_2$ is hard to characterize. Indeed the original ElasticNet paper [47] suggests using the heuristic of grid search to learn a good $\lambda_2$, even though $\lambda_1$ may be exactly optimized by computing full solution paths (for each $\lambda_2$). We approach this indirectly by utilizing a

---

[2]The soft-O notation is used to emphasize dependence on $T$, and suppresses other factors as well as logarithmic terms.

characterization of the LASSO solution by [42], which is based on the KKT (Karush–Kuhn–Tucker) optimality conditions, to arrive at a precise piecewise structure for the problem. In more detail, we use these conditions to come up with a set of algebraic curves (polynomial equations in $\lambda_1$ and $\lambda_2$) of bounded degrees, such that the set of possible discontinuities is contained within the zero-set of these curves, and the loss function behaves well in the each piece of the partition of the parameter domain by these curves. This characterization is crucial in establishing a bound on the structural complexity needed to provide strong generalization guarantees. We further show additional structure on these algebraic curves that (roughly speaking) imply that the curves do not concentrate in any region of the domain, allowing us to use the powerful recipe of [8] for online learning.

**Related work**. Model selection for Ridge regression, LASSO or ElasticNet typically involves selecting the regularization parameter $\lambda$ for given data, although some parameter-free techniques for variable selection have been recently proposed [32]. Choosing 'optimal' parameters for tuning the regularization has been a subject of extensive theoretical and applied research. Much of this effort is heuristic [26, 31] or focused on developing tuning objectives beyond validation accuracy like AIC or BIC [1, 39] without providing procedures for provably optimizing them. The standard approach given a tuning objective is to optimize it over a grid or random set of parameters, for which there are guarantees [17], but this does not ensure optimality over the entire continuous tuning domain, especially since objectives such as 0-1 validation error or information criteria can have many discontinuities. Selecting a grid that is too fine or too coarse can result in either very inefficient or highly inaccurate estimates (respectively) for good parameters. Other guarantees make strong assumptions on the data distribution such as sub-Gaussian noise [44, 16] or depend on unknown parameters that are hard to quantify in practice [23]. Recent work has shown asymptotic consistency of cross-validation for ridge regression, even in the limiting case $\lambda_2 \to 0$ which is particularly interesting for the overparameterized regime [29, 36]. A successful line of work has focused on efficiently obtaining models for different values of $\lambda_1$ using regularization paths [22], but the guarantees are computational rather than learning-theoretic or statistical. In contrast, we provide principled approaches that guarantee near-optimality of selected parameters with high confidence over the entire continuous domain of parameters.

Data-driven algorithm design has proved successful for tuning parameters for a variety of combinatorial problems like clustering, integer programming, auction design and graph-based learning [7, 11, 5, 4]. We provide an application of these techniques to parameter tuning in a problem that is not inherently combinatorial by revealing a novel discrete structure. We identify the underlying piecewise structure of the ElasticNet loss function which is extremely effective in establishing learning-theoretic guarantees [10]. To exploit this piecewise structure, we analyze the learning-theoretic complexity of rational algebraic function classes and infer generalization guarantees. We also employ and extend general tools and techniques for online data-driven learning from [8, 4] to rational functions in order to prove our online learning guarantees for regularization coefficient tuning.

## 2   Preliminaries and a Key Structural Result

Given data $(X, y)$ with $X \in \mathbb{R}^{m \times p}$ and $y \in \mathbb{R}^m$, consisting of $m$ labeled examples with $p$ features, we seek estimators $\beta \in \mathbb{R}^p$ which minimize the regularized loss. Popular regularization methods like LASSO and ElasticNet can be expressed as computing the solution of an optimization problem

$$\hat{\beta}_{\lambda,f}^{(X,y)} \in \arg\min_{\beta \in \mathbb{R}^p} \|y - X\beta\|_2^2 + \langle \lambda, f(\beta) \rangle$$

where $f : \mathbb{R}^p \to \mathbb{R}_{\geq 0}^d$ gives the regularization penalty for estimator $\beta$, $\lambda \in \mathbb{R}_{\geq 0}^d$ is the regularization parameter, and $d$ is the number of regularization parameters. $d = 1$ for Ridge and LASSO, and $d = 2$ for the ElasticNet. Setting $f = f_2$ with $f_2(\beta) = \|\beta\|_2^2$ yields Ridge regression, and setting $f(\beta) = f_1(\beta) := \|\beta\|_1$ corresponds to LASSO. Also using $f_{\text{EN}}(\beta) = (f_1(\beta), f_2(\beta))$ gives the ElasticNet with regularization parameter $\lambda = (\lambda_1, \lambda_2)$. Note that we use the same $\lambda$ (with some notational overloading) to denote the regularization parameters for ridge, LASSO, or ElasticNet. We write $\hat{\beta}_{\lambda,f}^{(X,y)}$ as simply $\hat{\beta}_{\lambda,f}$ when the dataset $(X, y)$ is clear from context. On any instance $x \in \mathbb{R}^p$ from the feature space, the prediction of the regularized estimator is given by the dot product $\langle x, \hat{\beta}_{\lambda,f} \rangle$. The average squared loss over a dataset $(X', y')$ with $X' \in \mathbb{R}^{m' \times p}$ and $y' \in \mathbb{R}^{m'}$ is given

by $l_r(\hat{\beta}_{\lambda,f}, (X', y')) = \frac{1}{m'} \left\| y' - X'\hat{\beta}_{\lambda,f} \right\|_2^2$. By setting $(X', y')$ to be the training data $(X, y)$, we get the training loss $l_r(\hat{\beta}_{\lambda,f}, (X, y))$. We use $(X_{\text{val}}, y_{\text{val}})$ to denote a validation split.

*Distributional and Online Settings.* In the *distributional or statistical* setting, we receive a collection of $n$ instances of the regression problem $P^{(i)} = (X^{(i)}, y^{(i)}, X_{\text{val}}^{(i)}, y_{\text{val}}^{(i)}) \in \mathcal{R}_{m_i, p_i, m_i'} :=$ $\mathbb{R}^{m_i \times p_i} \times \mathbb{R}^{m_i} \times \mathbb{R}^{m_i' \times p_i} \times \mathbb{R}^{m_i'}$ for $i \in [n]$ generated i.i.d. from some problem distribution $\mathcal{D}$. The problems are in the problem space given by $\Pi_{m,p} = \bigcup_{m_1 \geq 0, m_2 \leq m, p_1 \leq p} \mathcal{R}_{m_1, p_1, m_2}$ (note that the problem distribution $\mathcal{D}$ is over $\Pi_{m,p}$). On any given instance $P^{(i)}$ the loss is given by the squared loss on the validation set, $\ell_{\text{EN}}(\lambda, P^{(i)}) = l_r(\hat{\beta}_{\lambda, f_{\text{EN}}}^{(X^{(i)}, y^{(i)})}, (X_{\text{val}}^{(i)}, y_{\text{val}}^{(i)}))$. On the other hand, in the *online setting*, we receive a sequence of $T$ instances of the ElasticNet regression problem $P^{(i)} = (X^{(i)}, y^{(i)}, X_{\text{val}}^{(i)}, y_{\text{val}}^{(i)}) \in \Pi_{m,p}$ for $i \in [T]$ online. On any given instance $P^{(i)}$, the online learner is required to select the regularization parameter $\lambda^{(i)}$ without observing $y_{\text{val}}^{(i)}$, and experiences loss given by $\ell(\lambda^{(i)}, P^{(i)}) = l_c(\hat{\beta}_{\lambda^{(i)}, f_{EN}}^{(X^{(i)}, y^{(i)})}, (X_{\text{val}}^{(i)}, y_{\text{val}}^{(i)}))$. The goal is to minimize the regret w.r.t. choosing the best fixed parameter in hindsight for the same problem sequence, i.e. $R_T = \sum_{i=1}^T \ell(\lambda^{(i)}, P^{(i)}) - \min_\lambda \sum_{i=1}^T \ell(\lambda, P^{(i)})$. We also define average regret as $\frac{1}{T} R_T$ and expected regret as $\mathbb{E}[R_T]$ where the expectation is over both the randomness of the loss functions and any random coins used by the online algorithm.

Given a class of regularization algorithms $\mathcal{A}$ parameterized by regularization parameter $\lambda$ over a set of problem instances $\mathcal{X}$, and given loss function $\ell : \mathcal{A} \times \mathcal{X} \to \mathbb{R}$ which measures the loss of any algorithm in $\mathcal{A}$ on any fixed problem instance, consider the set of functions $\mathcal{H}_\mathcal{A} = \{\ell(A, \cdot) \mid A \in \mathcal{A}\}$. For example, for the ElasticNet we have $\ell_{\text{EN}}(\lambda, P) = l_r(\hat{\beta}_{\lambda, f_{\text{EN}}}^{(X_P, y_P)}, (X_P', y_P'))$, where $(X_P, y_P)$ and $(X_P', y_P')$ are the training and validation sets associated with problem $P \in \mathcal{X}$ respectively. Bounding the pseudo-dimension of $\mathcal{H}_\mathcal{A}$ gives a bound on the sample complexity for uniform convergence guarantees, i.e. a bound on the sample size $n$ for which the algorithm $\hat{A}_S \in \mathcal{A}$ which minimizes the average loss on any sample $S$ of size $n$ drawn i.i.d. from any problem distribution $\mathcal{D}$ is guaranteed to be near-optimal with high probability [21]. See Appendix A for the relevant classic definitions and results. Define the *dual class* $\mathcal{H}^*$ of a set of real-valued functions $\mathcal{H} \subseteq 2^\mathcal{X}$ as $\mathcal{H}^* = \{h_x^* : \mathcal{H} \to \mathbb{R} \mid x \in \mathcal{X}\}$ where $h_x^*(h) = h(x)$. In the context of regression problems $\mathcal{X}$, for each fixed problem instance $x \in \mathcal{X}$ there is a dual function $h_x^*$ that computes the loss $\ell(A, x)$ for any (primal) function $h_A = \ell(A, \cdot) \in \mathcal{H}_\mathcal{A}$. For a function class $\mathcal{H}$, showing that dual class $\mathcal{H}^*$ is piecewise-structured in the sense of Definition 1 and bounding the complexity of the duals of boundary and piece functions of $\mathcal{H}^*$ are useful to understand the learnability of $\mathcal{H}$ [10].

**Definition 1** (Piecewise structured functions, [10]). *A function class $H \subseteq \mathbb{R}^\mathcal{X}$ that maps a domain $\mathcal{X}$ to $\mathbb{R}$ is $(F, G, k)$-piecewise decomposable for a class $G \subseteq \{0, 1\}^\mathcal{X}$ of boundary functions and a class $F \subseteq \mathbb{R}^\mathcal{X}$ of piece functions if the following holds: for every $h \in H$, there are $k$ boundary functions $g_1, \ldots, g_k \in G$ and a piece function $f_\mathbf{b} \in F$ for each bit vector $\mathbf{b} \in \{0, 1\}^k$ such that for all $x \in \mathcal{X}$, $h(x) = f_{\mathbf{b}_x}(x)$ where $\mathbf{b}_x = (g_1(x), \ldots, g_k(x)) \in \{0, 1\}^k$.*

Intuitively, a real-valued function is piecewise-structured if the domain can be divided into pieces by a finite number of boundary functions (say linear or polynomial thresholds) and the function value over each piece is easy to characterize (e.g. constant, linear, polynomial). To state and understand our structural insights into the ElasticNet problem we will also need the definition of equicorrelation sets, the subset of features with maximum absolute correlation for any fixed $\lambda_1$, useful for characterizing LASSO/ElasticNet solutions. For any subset $\mathcal{E} \subseteq [p]$ of the features, we define $X_\mathcal{E} = (\ldots X_{*i} \ldots)_{i \in \mathcal{E}}$ as the $m \times |\mathcal{E}|$ matrix of columns $X_{*i}$ of $X$ corresponding to indices $i \in \mathcal{E}$. Similarly $\beta_\mathcal{E} \in \mathbb{R}^{|\mathcal{E}|}$ is the subset of estimators in $\beta$ corresponding to indices in $\mathcal{E}$. We will assume all the feature matrixes $X$ (for training datasets) are in general position (Definition 6).

**Definition 2** (Equicorrelation sets, [42]). *Let $\beta^* \in \arg\min_{\beta \in \mathbb{R}^p} \|y - X\beta\|_2^2 + \lambda_1 ||\beta||_1$. The equicorrelation set corresponding to $\beta^*$, $\mathcal{E} = \{j \in [p] \mid |\mathbf{x}_j^T(y - X\beta^*)| = \lambda_1\}$, is simply the set of covariates with maximum absolute correlation. We also define the equicorrelation sign vector for $\beta^*$ as $s = \mathsf{sign}(X_\mathcal{E}^T(y - X\beta^*)) \in \{\pm 1\}$.*

Consider the class of algorithms consisting of ElasticNet regressors for different values of $\lambda = (\lambda_1, \lambda_2) \in (0, \infty) \times (0, \infty)$. We assume $\lambda_1 > 0$ for technical simplicity (cf. [42]). We seek to solve

problems of the form $P = (X, y, X_{\text{val}}, y_{\text{val}}) \in \Pi_{m,p}$, where $(X, y)$ is the training set, $(X_{\text{val}}, y_{\text{val}})$ is the validation set with the same set of features, and $m, p$ are upper bounds on the number of examples and features respectively in any dataset. Let $\mathcal{H}_{\text{EN}} = \{\ell_{\text{EN}}(\lambda, \cdot) \mid \lambda \in (0, \infty) \times (0, \infty)\}$ denote the set of loss functions for the class of algorithms consisting of ElasticNet regressors for different values of $\lambda \in \mathbb{R}^+ \times \mathbb{R}^+$. Additionally, we will consider information criterion based loss functions, $\ell_{\text{EN}}^{\text{AIC}}(\lambda, P) = \ell_{\text{EN}}(\lambda, P) + 2||\hat{\beta}_{\lambda, f_{\text{EN}}}^{(X,y)}||_0$ and $\ell_{\text{EN}}^{\text{BIC}}(\lambda, P) = \ell_{\text{EN}}(\lambda, P) + 2||\hat{\beta}_{\lambda, f_{\text{EN}}}^{(X,y)}||_0 \log m$ [1, 39]. Let $\mathcal{H}_{\text{EN}}^{\text{AIC}}$ and $\mathcal{H}_{\text{EN}}^{\text{BIC}}$ denote the corresponding sets of loss functions. These criteria are popularly used to compute the squared loss on the training set, to give alternatives to cross-validation. We do not make any assumption on the relation between training and validation sets in our formulation, so our analysis can capture these settings as well.

We will now establish a piecewise structure of the dual class loss functions (Definition 1). A key observation is that if the signed equicorrelation set $(\mathcal{E}, s)$ (i.e. a subset of features $\mathcal{E} \subseteq [p]$ with the same maximum absolute correlation, assigned a fixed sign pattern $\{-1, +1\}^{|\mathcal{E}|}$, see Definition 2) is fixed, then the ElasticNet coefficients may be characterized (Lemma C.1) and the loss is a fixed rational polynomial piece function of the parameters $\lambda_1, \lambda_2$. We then show the existence of a set of boundary function curves $\mathcal{G}$, such that any region of the parameter space located on a fixed side of all the curves (more formally, for a fixed sign pattern in Definition 1) in $\mathcal{G}$ has the same signed equicorrelation set. The boundary functions are a collection of possible curves at which a covariate may enter or leave the set $\mathcal{E}$ and correspond to algebraic curves. We make repeated use of the following lemma which provides useful properties of the piece functions as well the the boundary functions of the dual class loss functions.

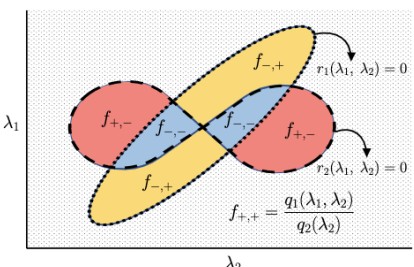

Figure 1: An illustration of the piecewise structure of the ElasticNet loss, as a function of the regularization parameters, for a fixed problem instance. Pieces are regions where some bounded degree polynomials $(r_1, r_2)$ have a fixed sign pattern (one of $\pm 1, \pm 1$), and in each piece the loss is a fixed (rational) function.

**Lemma 2.1.** *Let $A$ be an $r \times s$ matrix. Consider the matrix $B(\lambda) = (A^T A + \lambda I_s)^{-1}$ and $\lambda > 0$.*

1. *Each entry of $B(\lambda)$ is a rational polynomial $P_{ij}(\lambda)/Q(\lambda)$ for $i, j \in [s]$ with each $P_{ij}$ of degree at most $s - 1$, and $Q$ of degree $s$.*

2. *Further, for $i = j$, $P_{ij}$ has degree $s - 1$ and leading coefficient 1, and for $i \neq j$ $P_{ij}$ has degree at most $s - 2$. Also, $Q(\lambda)$ has leading coefficient 1.*

The proof is straightforward (Appendix C). We will now formally state and prove our key structural result which is needed to establish our generalization and online regret guarantees in Section 3.

**Theorem 2.2.** *Let $\mathcal{L}$ be a set of functions $\{l_\lambda : \Pi_{m,p} \to \mathbb{R}_{\geq 0} \mid \lambda \in \mathbb{R}^+ \times \mathbb{R}_{\geq 0}\}$ that map a regression problem instance $P \in \Pi_{m,p}$ to the validation loss $\ell_{\text{EN}}(\lambda, P)$ of ElasticNet trained with regularization parameter $\lambda = (\lambda_1, \lambda_2)$. The dual class $\mathcal{L}^*$ is $(\mathcal{F}, \mathcal{G}, p3^p)$-piecewise decomposable, with $\mathcal{F} = \{f_q : \mathcal{L} \to \mathbb{R}\}$ consisting of rational polynomial functions $f_{q_1, q_2} : l_\lambda \mapsto \frac{q_1(\lambda_1, \lambda_2)}{q_2(\lambda_2)}$, where $q_1, q_2$ have degrees at most $2p$, and $\mathcal{G} = \{g_r : \mathcal{L} \to \{0, 1\}\}$ consisting of semi-algebraic sets[3] bounded by algebraic curves $g_r : u_\lambda \mapsto \mathbb{I}\{r(\lambda_1, \lambda_2) < 0\}$, where $r$ is a polynomial of degree 1 in $\lambda_1$ and at most $p$ in $\lambda_2$.*

*Proof.* Let $P = (X, y, X_{\text{val}}, y_{\text{val}}) \in \Pi_{m,p}$ be a regression problem instance. By using the standard reduction to LASSO [47] and well-known characterization of the LASSO solution in terms of equicorrelation sets, we can characterize the solution $\hat{\beta}_{\lambda, f_{EN}}$ of the Elastic Net as follows (Lemma C.1):

$$\hat{\beta}_{\lambda, f_{EN}} = (X_\mathcal{E}^T X_\mathcal{E} + \lambda_2 I_{|\mathcal{E}|})^{-1} X_\mathcal{E}^T y - \lambda_1 (X_\mathcal{E}^T X_\mathcal{E} + \lambda_2 I_{|\mathcal{E}|})^{-1} s$$

for some $\mathcal{E} \in [p]$ and $s \in \{-1, 1\}^p$. Thus for any $\lambda = (\lambda_1, \lambda_2)$, the prediction $\hat{y}$ on any validation example with features $\boldsymbol{x} \in \mathbb{R}^p$ satisfies (for some $\mathcal{E}, s \in 2^{[p]} \times \{-1, 1\}^p$)

$$\hat{y}_j = \boldsymbol{x}\hat{\beta}_{\lambda, f_{EN}} = \boldsymbol{x}(X_\mathcal{E}^T X_\mathcal{E} + \lambda_2 I_{|\mathcal{E}|})^{-1} X_\mathcal{E}^T y - \lambda_1 \boldsymbol{x}(X_\mathcal{E}^T X_\mathcal{E} + \lambda_2 I_{|\mathcal{E}|})^{-1} s$$

---

[3] See Definition 7 for definitions of standard terminology from algebraic geometry.

For any subset $R \subseteq \mathbb{R}^2$, if the signed equicorrelation set $(\mathcal{E}, s)$ is fixed over $R$, then the above observation, together with Lemma C.2 implies that the loss function $\ell_{\mathsf{EN}}(\lambda, P)$ is a rational function of the form $\frac{q_1(\lambda_1, \lambda_2)}{q_2(\lambda_2)}$, where $q_1$ is a bivariate polynomial with degree at most $2|\mathcal{E}|$ and $q_2$ is univariate with degree $2|\mathcal{E}|$.

To show the piecewise structure, we need to demonstrate a set boundary functions $\mathcal{G} = \{g_1, \ldots, g_k\}$ such that for any sign pattern $\mathbf{b} \in \{0, 1\}^k$, the signed equicorrelation set $(\mathcal{E}, s)$ for the region with sign pattern $\mathbf{b}$ is fixed. To this end, based on the observation above, we will consider the conditions (on $\lambda$) under which a covariate may enter or leave the equicorrelation set. We will show that this can happen only at one of a finite number of algebraic curves (with bounded degrees).

*Condition for joining $\mathcal{E}$.* Fix $\mathcal{E}, s$. Also fix $j \notin \mathcal{E}$. If covariate $j$ enters the equicorrelation set, the KKT conditions (Lemma B.1) applied to the LASSO problem corresponding to the ElasticNet (Lemma C.1) imply

$$(\boldsymbol{x}_j^*)^T(y^* - X_{\mathcal{E}}^*(c_1 - c_2\lambda_1^*)) = \pm\lambda_1^*,$$

where $c_1 = (X_{\mathcal{E}}^{*T}X_{\mathcal{E}}^*)^{-1}X_{\mathcal{E}}^{*T}y^*$, $c_2 = (X_{\mathcal{E}}^{*T}X_{\mathcal{E}}^*)^{-1}s$, $X^* = \frac{1}{\sqrt{1+\lambda_2}}\begin{pmatrix} X \\ \sqrt{\lambda_2}I_p \end{pmatrix}$, $y^* = \begin{pmatrix} y \\ 0 \end{pmatrix}$, and $\lambda_1^* = \frac{\lambda_1}{\sqrt{1+\lambda_2}}$. Rearranging, and simplifying, we get

$$\lambda_1^* = \frac{(\boldsymbol{x}_j^*)^T X_{\mathcal{E}}^*(X_{\mathcal{E}}^{*T}X_{\mathcal{E}}^*)^{-1}(X_{\mathcal{E}}^*)^T y^* - (\boldsymbol{x}_j^*)^T y^*}{(\boldsymbol{x}_j^*)^T X_{\mathcal{E}}^*(X_{\mathcal{E}}^{*T}X_{\mathcal{E}}^*)^{-1}s \pm 1}, \text{ or}$$

$$\lambda_1 = \frac{\boldsymbol{x}_j^T X_{\mathcal{E}}(X_{\mathcal{E}}^T X_{\mathcal{E}} + \lambda_2 I_{|\mathcal{E}|})^{-1}X_{\mathcal{E}}^T y - \boldsymbol{x}_j^T y}{\boldsymbol{x}_j^T X_{\mathcal{E}}(X_{\mathcal{E}}^T X_{\mathcal{E}} + \lambda_2 I_{|\mathcal{E}|})^{-1}s \pm 1}.$$

Note that the terms $(\boldsymbol{x}_j^*)^T X_{\mathcal{E}}^* = \boldsymbol{x}_j^T X_{\mathcal{E}}$, $(X_{\mathcal{E}}^*)^T y^* = X_{\mathcal{E}}^T y$, and $(\boldsymbol{x}_j^*)^T y^* = \boldsymbol{x}_j^T y$ do not depend on $\lambda_1$ or $\lambda_2$ (the $\lambda_2$ terms are zeroed out since $j \notin \mathcal{E}$). Moreover, $(X_{\mathcal{E}}^{*T}X_{\mathcal{E}}^*)^{-1} = (X_{\mathcal{E}}^T X_{\mathcal{E}} + \lambda_2 I_{|\mathcal{E}|})^{-1}$. Using Lemma C.2, we get an algebraic curve $r_{j,\mathcal{E},s}(\lambda_1, \lambda_2) = 0$ with degree 1 in $\lambda_1$ and $|\mathcal{E}|$ in $\lambda_2$ corresponding to addition of $j \notin \mathcal{E}$ given $\mathcal{E}, s$.

*Condition for leaving $\mathcal{E}$.* Now consider a fixed $j' \in \mathcal{E}$, given fixed $\mathcal{E}, s$. The coefficient of $j'$ will be zero for $\lambda_1^* = \frac{(c_1)_{j'}}{(c_2)_{j'}}$, which simplifies to $\lambda_1((X_{\mathcal{E}}^T X_{\mathcal{E}} + \lambda_2 I_{|\mathcal{E}|})^{-1}s)_{j'} = ((X_{\mathcal{E}}^T X_{\mathcal{E}} + \lambda_2 I_{|\mathcal{E}|})^{-1}X_{\mathcal{E}}^T y)_{j'}$. Again by Lemma C.2, we get an algebraic curve $r_{j',\mathcal{E},s}(\lambda_1, \lambda_2) = 0$ with degree 1 in $\lambda_1$ and at most $|\mathcal{E}|$ in $\lambda_2$ corresponding to removal of $j' \in \mathcal{E}$ given $\mathcal{E}, s$.

Putting the two together, we get $\sum_{i=0}^{p} 2^i \binom{p}{i}((p-i)+i) = p3^p$ algebraic curves of degree 1 in $\lambda_1$ and at most $p$ in $\lambda_2$, across which the signed equicorrelation set may change. These curves characterize the complete set of points $(\lambda_1, \lambda_2)$ at which $(\mathcal{E}, s)$ may possibly change. Thus by setting these $p3^p$ curves as the set of boundary functions $\mathcal{G}$, $\mathcal{E}, s$ is guaranteed to be fixed for each sign pattern, and the corresponding loss takes the rational function form shown above. $\qquad\square$

The exact same piecewise structure can be established for the dual function classes for loss functions $\ell_{\mathsf{EN}}^{\mathsf{AIC}}(\lambda, \cdot)$ and $\ell_{\mathsf{EN}}^{\mathsf{BIC}}(\lambda, \cdot)$. This is evident from the proof of Theorem 2.2, since any dual piece has a fixed equicorrelation set, and therefore $||\beta||_0$ is fixed. Given this piecewise structure, a challenge to learning values of $\lambda$ that minimize the loss function is that the function may not be differentiable (or may even be discontinuous, for the information criteria based losses) at the piece boundaries, making well-known gradient-based (local) optimization techniques inapplicable here. In the following (specifically Algorithm 1) we will show that techniques from data-driven design may be used to overcome this optimization challenge.

## 3 Learning to Regularize the ElasticNet

We will consider the problem of learning provably good ElasticNet parameters for a given problem domain, from multiple datasets (problem instances) either available as a collection (Section 3.1), or arriving online (Section 3.2). Our parameter tuning techniques also apply to simpler regression techniques typically used for variable selection, like LARS and LASSO, which are reasonable choices if the features are not multicollinear. Proof details for this section are located in Appendix C.

## 3.1 Distributional Setting

Our main result in this section is the following upper bound on the pseudo-dimension of the classes of loss functions for the ElasticNet, which implies that in our distributional setting it is possible to learn near-optimal values of $\lambda$ with polynomially many problem instances.

**Theorem 3.1.** $\text{PDIM}(\mathcal{H}_{EN}) = O(p^2)$. *Further,* $\text{PDIM}(\mathcal{H}_{EN}^{AIC}) = O(p^2)$ *and* $\text{PDIM}(\mathcal{H}_{EN}^{BIC}) = O(p^2)$.

*Proof Sketch.* We use the $(\mathcal{F}, \mathcal{G}, p3^p)$-piecewise decomposable structure for the dual class function $\mathcal{H}_{EN}^*$ established in Theorem 2.2. We can bound the pseudo-dimension of the dual class of piece functions $\mathcal{F}^*$ (a class of bivariate rational functions) by $O(\log p)$ by giving an upper bound (of $O(k^3 d^3)$) on the number of sign patterns over $\mathbb{R}^2$ induced by $k$ algebraic curves of degree at most $d$. We can also bound the VC dimension of the dual class of boundary functions $\mathcal{G}^*$ (semi-algebraic sets in two variates) by $O(p)$ using a standard linearization argument. Finally, a powerful result from [10] (Theorem C.3) allows us to bound the pseudodimension of $\mathcal{H}$ by combining the above results. $\square$

A key challenge to establish the theorem is providing new bounds on the pseudo-dimension of rational functions of bounded degrees (Lemma C.5). The upper bound above implies a guarantee on the sample complexity of learning the ElasticNet tuning parameter, using standard learning-theoretic results [2]. In our setting of learning from multiple problem instances, each sample is a dataset instance, so the sample complexity is simply the number of regression problem instances needed to learn the tuning parameters to any given approximation and confidence level.

**Theorem 3.2** (Sample complexity of tuning the ElasticNet). *Let $\mathcal{D}$ be an arbitary distribution over the problem space $\Pi_{m,p}$. There is an algorithm which given $n = O\left(\frac{1}{\epsilon^2}(p^2 \log \frac{1}{\epsilon} + \log \frac{1}{\delta})\right)$ problem samples drawn from $\mathcal{D}$, for any $\epsilon > 0$ and $\delta \in (0,1)$, outputs a regularization parameter $\hat{\lambda}$ for the ElasticNet such that with probability at least $1 - \delta$ over the draw of the problem samples, we have that*

$$\left| \mathbb{E}_{P \sim \mathcal{D}}[\ell_{EN}(\hat{\lambda}, P)] - \min_\lambda \mathbb{E}_{P \sim \mathcal{D}}[\ell_{EN}(\lambda, P)] \right| \leq \epsilon$$

*Proof.* This follows from substituting our result in Theorem 3.1 into well-known generalization guarantee for function classes with bounded pseudo-dimensions (Theorem A.1). $\square$

*Discussion and applications.* Computing the parameters which minimize the loss on the problem samples (aka Empirical Risk Minimization, or ERM) achieves the sample complexity bound in Theorem 3.2. Even though we only need polynomially many samples to guarantee the selection of nearly-optimal parameters, it is not clear how to implement the ERM efficiently. Note that we do not assume the set of features is the same across problem instances, so our approach can handle feature reset i.e. different problem instances can differ in not only the number of examples but also the number of features. Moreover, as a special case application, we consider the commonly used techniques of leave-one-out cross validation (LOOCV) and Monte Carlo cross validation (repeated random test-validation splits, typically independent and in a fixed proportion). Given a dataset of size $m_{tr}$, LOOCV would require $m_{tr}$ regression fits which can be inefficient for large dataset size. Alternately, we can consider draws from a distribution $\mathcal{D}_{LOO}$ which generates problem instances $P$ from a fixed dataset $(X, y) \in \mathbb{R}^{m+1 \times p} \times \mathbb{R}^{m+1}$ by uniformly selecting $j \in [m+1]$ and setting $P = (X_{-j*}, y_{-j}, X_{j*}, y_j)$. Theorem 3.2 now implies that $\tilde{O}(p^2/\epsilon^2)$ iterations are enough to determine an ElasticNet parameter $\hat{\lambda}$ with loss within $\epsilon$ (w.h.p.) of the parameter $\lambda^*$ obtained from running the full LOOCV. Similarly, we can define a distribution $\mathcal{D}_{MC}$ to capture the Monte Carlo cross validation procedure and determine the number of iterations sufficient to get an $\epsilon$-approximation of the loss corresponding parameter selection with arbitrarily large number of runs of the procedure. Thus, in a very precise sense, our results answer the question of how much cross-validation is enough to effectively implement the above techniques.

**Remark 1.** *While our result implies polynomial sample complexity, the question of learning the provably near-optimal parameter efficiently (even in output polynomial time) is left open. For the special cases of LASSO ($\lambda_2 = 0$) and Ridge ($\lambda_1 = 0$), the piece boundaries of the piecewise polynomial dual class (loss) function may be computed efficiently (using the LARS-LASSO algorithm of [22] for LASSO, and solving linear systems and locating roots of polynomials for Ridge). This applies to online and classification settings in the following sections as well.*

## 3.2 Online Learning

We will now extend our results to learning the regularization coefficients given an online sequence of regression problems, such as when one needs to solve a new regression problem each day. Unlike the distributional setting above, we will not assume any problem distribution and our results will hold for an adversarial sequence of problem instances. We will need very mild assumptions on the data, namely boundedness of feature and prediction values and 'smoothness' of predictions (formally stated as Assumptions 1 and 2), while our distributional results above hold for worst-case problem datasets. Our first assumption is that all feature values and predictions are bounded, for training as well as validation examples.

**Assumption 1** (Boundedness). *The predicted variable and all feature values are bounded by an absolute constant R, i.e.* $\max\{||X^{(i)}||_{\infty,\infty}, ||y^{(i)}||_{\infty}, ||X_{val}^{(i)}||_{\infty,\infty}, ||y_{val}^{(i)}||_{\infty}\} \leq R$.

We will need the following definition of distribution smoothness to state our second assumption.

**Definition 3.** *A continuous probability distribution is said to be $\kappa$-bounded if the probability density function $p(x)$ satisfies $p(x) \leq \kappa$ for any $x$ in the sample space.*

For example, the *normal* distribution $\mathcal{N}(\mu, \sigma^2)$ with mean $\mu$ and standard deviation $\sigma$ is $\frac{1}{\sigma\sqrt{2\pi}}$-bounded. We assume that the predicted variable $y$ in the training set comes from a $\kappa$-bounded (i.e. smooth) distribution, which does not require the strong tail decay of sub-Gaussian distributions [44, 13]. Moreover, the online adversary is allowed to change the distribution as long as it is $\kappa$-bounded. Note that our assumption also captures common data preprocessing steps, for example the jitter parameter in the popular Python library scikit-learn [37] adds a uniform noise to the $y$ values to help model stability. The assumption is formally stated as follows:

**Assumption 2** (Smooth predictions). *The predicted variables $y^{(i)}$ in the training set are drawn from a joint $\kappa$-bounded distribution, i.e. for each $i$, the variables $y^{(i)}$ have a joint distribution with probability density bounded by $\kappa$.*

Under these assumptions, we can show that it is possible to learn the ElasticNet parameters with sublinear expected regret when the problem instances arrive online. The learning algorithm (Algorithm 1) that achieves this regret is a continuous variant of the classic Exponential Weights algorithm [14, 6]. It samples points in the domain with probability inversely propotional to the exponentiated loss. To formally state our result, we will need the following definition of *dispersed* loss functions. Informally speaking, it captures how amenable a set of non-Lipschitz functions is to online learning by measuring the worst rate of occurrence of non-Lipschitzness (or discontinuities) between any pair of points in the domain. [6, 9, 8] show that dispersion is necessary and sufficient for learning piecewise Lipschitz functions.

**Definition 4.** *Dispersion [8]. The sequence of random loss functions $l_1, \ldots, l_T$ is $\beta$-dispersed for the Lipschitz constant $L$ if, for all $T$ and for all $\epsilon \geq T^{-\beta}$, we have that, in expectation, at most $\tilde{O}(\epsilon T)$ functions (the soft-O notation suppresses dependence on quantities beside $\epsilon, T$ and $\beta$, as well as logarithmic terms) are not $L$-Lipschitz for any pair of points at distance $\epsilon$ in the domain $\mathcal{C}$. That is, for all $T$ and for all $\epsilon \geq T^{-\beta}$, $\mathbb{E}\left[\max_{\substack{\rho,\rho' \in \mathcal{C} \\ ||\rho-\rho'||_2 \leq \epsilon}} \left|\{t \in [T] \mid l_t(\rho) - l_t(\rho') > L \|\rho - \rho'\|_2\}\right|\right] \leq \tilde{O}(\epsilon T)$.*

Our key contribution is to show that the loss sequence is dispersed (Definition 4) under the above assumptions. This involves establishing additional structure for the problem, specifically about the location of boundary functions in the piecewise structure from Theorem 2.2. This stronger characterization coupled with results from [8] on dispersion of algebraic discontinuities completes the proof.

**Theorem 3.3.** *Suppose Assumptions 1 and 2 hold. Let $l_1, \ldots, l_T : (0, \lambda_{\max})^2 \to \mathbb{R}_{\geq 0}$ denote an independent sequence of losses (e.g. fresh randomness is used to generate the validation set features in each round) as a function of the ElasticNet regularization parameter $\lambda = (\lambda_1, \lambda_2)$, $l_i(\lambda) = l_r(\hat{\beta}_{\lambda, f_{EN}}^{(X^{(i)}, y^{(i)})}, (X_{val}^{(i)}, y_{val}^{(i)}))$. The sequence of functions is $\frac{1}{2}$-dispersed, and there is an online algorithm with $\tilde{O}(\sqrt{T})$[4] expected regret. The result also holds for loss functions adjusted by information criteria AIC and BIC.*

*Proof Sketch.* We start with the $(\mathcal{F}, \mathcal{G}, p3^p)$-piecewise decomposable structure for the dual class function $\mathcal{H}_{EN}^*$ from Theorem 2.2. Observe that the rational piece functions in $\mathcal{F}$ do not introduce

---

[4]The $\tilde{O}(\cdot)$ notation hides dependence on logarithmic terms, as well as on quantities other than $T$.

---

**Algorithm 1** Data-driven Regularization $(\zeta)$

---

1: **Input:** Problems $(X^{(i)}, y^{(i)})$ and regularization penalty function $f$.
2: **Hyperparameter:** step size parameter $\zeta \in (0, 1]$.
3: **Output:** Parameters $(\lambda_i)_{i \in [T]} \in C$, $C \subset \mathbb{R}^+$ (LASSO/Ridge) or $C \subset \mathbb{R}^{+^2}$ (ElasticNet).
4: Set $w_1(\lambda) = 1$ for all $\lambda \in C$.
5: **for** $i = 1, 2, \ldots, T$ **do**
6:    $W_i := \int_C w_i(\lambda) d\lambda$.
7:    Sample $\lambda$ with probability $p_t(\lambda) = \frac{w_i(\lambda)}{W_i}$, output as $\lambda_i$.
8:    Compute average loss function $l_i(\lambda) = \frac{1}{|y^{(i)}|} l(\hat{\beta}_{\lambda,f}, (X^{(i)}, y^{(i)}))$.
9:    For each $\lambda \in C$, update weights $w_{i+1}(\lambda) = e^{\zeta(1 - l_i(\lambda))} w_i(\lambda)$.

---

any new discontinuities since the denominator polynomials do not have positive roots. For each of two types of boundary functions in $\mathcal{G}$ (corresponding to leaving/entering the equicorrelation set) we show that the discontinuities between any pair of points $\lambda, \lambda'$ lie along the roots of polynomials with non-leading coefficients bounded and smoothly distributed (bounded joint density). This allows us to use results from [8] to establish dispersion, and therefore online learnability. $\qquad\square$

We remark that the above result holds for arbitrary training features and validation sets in the problem sequence that satisfy our assumptions, in particular the loss functions are only assumed to be independent but not identically distributed. In contrast, the results in the previous section needed them to be drawn from the same distribution. Also the parameters need to be selected online, and cannot be changed for already seen instances. This setting captures interesting practical settings where the set of features (including feature dimensions) and the relevant training set (including training set size) may change over the online sequence. It is not clear how usual model selection techniques like cross-validation may be adapted to these challenging settings.

## 4 Extension to Regularized Least Squares Classification

Regression techniques can also be used to train binary classifiers by using an appropriate threshold on top of the regression estimate. Intuitively, regression learns a linear mapping which projects the datapoints onto a one-dimensional space, i.e. a real number, after which a threshold may be applied to classify the points. The use of thresholds to make discrete classifications adds discontinuities to the empirical loss function. Thus, in general, the classification setting is more challenging as it already includes the piecewise structure in the regression loss. We provide statistical and online learning guarantees for Ridge and LASSO. For the ElasticNet we present the extensions needed to the arguments from the previous sections to obtain results in the classification setting.

More formally, we will restrict $y$ to $\{0, 1\}^m$. The estimator $\hat{\beta}_{\lambda,f}$ is obtained as before, and the prediction on a test instance $x$ may be obtained by taking the sign of a thresholded regression estimate, $\mathsf{sign}(\langle x, \hat{\beta}_{\lambda,f} \rangle - \tau)$, where $\mathsf{sign} : \mathbb{R} \to \{0, 1\}$ maps $x \in \mathbb{R}$ to $\mathbb{I}\{x \geq 0\}$ and $\tau \in \mathbb{R}$ is the *threshold*. The threshold $\tau$ corresponds to the intercept or bias of the learned linear classifier, here we will treat it as a tunable hyperparameter (in addition to $\lambda_1, \lambda_2$)[5]. The average 0-1 loss over the dataset $(X, y)$ is given by $l_c(\hat{\beta}_{\lambda,f}, (X, y), \tau) = \frac{1}{m} \sum_{i=1}^m |y_i - \mathsf{sign}(\langle X_i, \hat{\beta}_{\lambda,f} \rangle - \tau)|$[6]. Proofs from this section are in Appendix D.

### 4.1 Distributional setting

The problem setting is the same as in Section 3.1, except that the labels $y$ are binary and we use threshold for prediction. We bound the pseudo-dimension for classification loss on these problem instances, which as before (c.f. Theorems 3.1 and 3.2) imply that polynomially many problem samples are sufficient to generalize well over the problem distribution $\mathcal{D}$. For Ridge and LASSO we

---

[5]We can still have a problem instance specific bias in $\beta$ using the standard trick of adding a unit feature to $X$, thus we generalize the common practice of using a fixed threshold. For example, the RidgeClassifier implementation in Python library scikit-learn 1.1.1 [37] assumes $y \in \{-1, +1\}^m$ and sets $\tau = 0$.

[6]Squared loss and 0-1 loss are identical in this setting.

upper bound the number of discontinuities of the piecewise constant classification loss by determining the values of $\lambda$ where any prediction changes.

**Theorem 4.1.** *Let $\mathcal{H}^c_{Ridge}$, $\mathcal{H}^c_{LASSO}$ and $\mathcal{H}^c_{EN}$ denote the set of loss functions for classification problems with at most $m$ examples and $p$ features, for linear classifiers regularized using Ridge, LASSO and ElasticNet regression respectively.*

*(i)* $\text{PDIM}(\mathcal{H}^c_{Ridge}) = O(\log mp)$

*(ii)* $\text{PDIM}(\mathcal{H}^c_{LASSO}) = O(p \log m)$. *Further, in the overparameterized regime ($p \gg m$), we have that* $\text{PDIM}(\mathcal{H}^c_{LASSO}) = O(m \log \frac{p}{m})$.

*(iii)* $\text{PDIM}(\mathcal{H}^c_{EN}) = O(p^2 + p \log m)$.

The key difference with the bound for the regression loss in Theorem 3.1 is the additional $O(p \log m)$ term which corresponds to discontinuities induced by the thresholding in the regression based classifiers. We can establish a structure similar to Theorem 2.2 in this case (Lemma D.1).

### 4.2 Online setting

As in Section 3.2, we can define an online learning setting for classification. Note that the smoothness of the predicted variable is not meaningful here, since $y$ is a binary vector. Instead we will assume that the validation examples have smooth feature values. Intuitively this means that small perturbations to the feature values does not meaningfully change the problem.

**Assumption 3** (Smooth validation features). *The feature values $(X^{(i)}_{val})_{jk}$ in the validation examples are drawn from a joint $\kappa$-bounded distribution.*

Under the assumption, we show that we can learn the regularization parameters online, for each of Ridge, LASSO and ElasticNet estimators. The proofs are straightforward extensions of the structural results developed in the previous sections, with minor technical changes to use the above validation set feature smoothness instead of Assumption 2, and are deferred to the appendix.

**Theorem 4.2.** *Suppose Assumptions 1 and 3 hold. Let $l_1, \ldots, l_T : (0, H]^d \times [-H, H] \to \mathbb{R}$ denote an independent sequence of losses as a function of the regularization parameter $\lambda$, $l_i(\lambda, \tau) = l_c(\hat{\beta}_{\lambda,f}, (X^{(i)}, y^{(i)}), \tau)$. If $f$ is given by $f_1$ (LASSO), $f_2$ (Ridge), or $f_{EN}$ (ElasticNet) then the sequence of functions is $\frac{1}{2}$-dispersed and there is an online algorithm with $\tilde{O}(\sqrt{T})$ expected regret.*

## 5 Conclusions and Future Work

We obtain a novel structural result for the ElasticNet loss as a function of the tuning parameters. Our characterization gives polynomial upper bounds for the sample complexity of learning the parameters from multiple instances coming from the same problem domain. For the ElasticNet we show generalization and online regret guarantees, but efficient implementation of the algorithms is an interesting question for further work. Also we show general learning-theoretic guarantees, i.e. without any significant restrictions on the data-generating distribution, in learning from multiple problems. The problems may be drawn i.i.d. from an arbitrary *problem* distribution, or even arrive in an online sequence but with some smoothness properties. It is unclear if such guarantees may be given for tuning parameters for the more standard setting of tuning a single training set. In this work we only give upper bounds on the sample complexity by bounding the pseudodimension, showing lower bounds is an interesting direction for future work.

### Acknowledgments

We thank Ryan Tibshirani for helpful discussions. This material is based on work supported by the National Science Foundation under grants CCF-1910321, IIS-1705121, IIS-1838017, IIS-1901403, IIS-2046613, IIS-2112471, and SES-1919453; the Defense Advanced Research Projects Agency under cooperative agreement HR00112020003; a Simons Investigator Award; an AWS Machine Learning Research Award; an Amazon Research Award; a Bloomberg Research Grant; a Microsoft Research Faculty Fellowship; funding from Meta, Morgan Stanley, and Amazon; and a Facebook PhD Fellowship. Any opinions, findings and conclusions or recommendations expressed in this material are those of the author(s) and do not necessarily reflect the views of any of these funding agencies.

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
