# OpenReview forum: "Provably tuning the ElasticNet across instances"
_NeurIPS.cc/2022/Conference — NeurIPS 2022 Accept_

### Official Review · Reviewer_zfht · 2022-06-27

**Rating:** 5
**Confidence:** 2
**Soundness:** 3 good
**Presentation:** 3 good
**Contribution:** 3 good

**Summary:**

The authors introduce a theoretical result that bounds the number of problems
needed to obtain a particular accuracy of $\lambda_1$ and $\lambda_2$ in
hyper-parameter optimization for the elastic net for both offline and online
learning.

**Questions:**

- ~The authors in [1] state that the regularization path has at worst $(3^p +
  1)/2$ segments. But you say, if I am understanding correctly, that it is
  exactly $3^p$ (l. 587). Perhaps I am missing something, but why is your result
  different? Should there be an O there?~
- ~Why is there only a proof sketch and not a full proof for theorem 3.1?~
- If I understand correctly, you arrive at your bound on the number of problems
  needed by assuming the worst-case situation in terms of path complexity. What
  is the impact of this on the tightness of the bound (and practical
  usefulness)? I would assume that this would make the bound conservative (since
  in practice no paths for real data contain $3^p$ segments).

**Limitations:**

- Consider demonstrating with a simple example what your results imply in terms
  of the number of problems you need to solve for some real data set. Also show
  via some simple experiment that standard approaches (such as 10-fold CV) can
  fail (as you claim they do) and that your method solves this problem.
- Move section 4 to the appendix since ordinary lasso classification is not a
  standard or useful model.
- ~State clearly what your contributions are in relation to previous work such as
  [1, 2, 3, 4] and cite their work where appropriate. In particular there
  seems to be overlap with [1] that isn't accounted for in the work.~
- ~Consider amending the literature section in the paper by including some or
  all of [1, 2, 3], and [4] regarding homotopy methods for the lasso.~
- Include at the very least [5] in your literature section on hyper-parameter
  tuning for the lasso.
- It is not standard to call the Elastic Net (or elastic net) ElasticNet.
  Consider adding a space.
- l. 58. There should be a period at the end of the sentence.
- l. 77. w.h.p. is not a standard abbreviation. Consider spelling it out.
- l. 208. There should be a period at the end of the equation.
- l. 271. There should be a period after $\epsilon$.
- l. 386. There is an extra space after "values".

[2] P. Garrigues and L. Ghaoui, “An homotopy algorithm for the lasso with online
    observations,” in Advances in neural information processing systems 21,
    Vancouver, Canada, Dec. 2008, vol. 21, pp. 489–496. [Online]. Available:
    https://proceedings.neurips.cc/paper/2008/file/38af86134b65d0f10fe33d30dd76442e-Paper.pdf

[3] D. M. Malioutov, M. Cetin, and A. S. Willsky, “Homotopy continuation for
    sparse signal representation,” in Proceedings. (ICASSP ’05). IEEE
    International Conference on Acoustics, Speech, and Signal Processing, 2005,
    Philadelphia, USA, Mar. 2005, vol. 5, pp. v733–v736. doi:
    10.1109/ICASSP.2005.1416408.

[4] M. Osborne, B. Presnell, and B. Turlach, “A new approach to variable
    selection in least squares problems,” IMA Journal of Numerical Analysis,
    vol. 20, no. 3, pp. 389–403, Jul. 2000, doi: 10.1093/imanum/20.3.389.

[5] Q. Bertrand, Q. Klopfenstein, M. Blondel, S. Vaiter, A. Gramfort, and J.
    Salmon, “Implicit differentiation of Lasso-type models for hyperparameter
    optimization,” in Proceedings of the 37th International Conference on
    Machine Learning, Nov. 2020, pp. 810–821. Accessed: Jun. 27, 2022. [Online].
    Available: https://proceedings.mlr.press/v119/bertrand20a.html

**Strengths And Weaknesses:**

- The paper seems to be theoretically sound and follows a logical progression.
- The literature section on exact (homotopy) methods for the elastic net and
  lasso is lacking and only insofar as I can see included the LARS paper at the
  moment.
- ~Section 2.1 and Lemma 2.1 seem to me to be related to the work in [1] and it is not quite clear what
  the novelty and enhancements of the current work are.~
- The bound on the number of problems needed for a certain accuracy in
  estimating $\lambda_1$ and $\lambda_2$ would be a useful addition, although it
  is not clear that the bound is useful in practice.
- Section 4 does not seem to address a useful problem. Classification with the
  ordinary (least-squares) elastic net is not a standard (or indeed useful) approach
  to classification as compared to elastic net-regularized logistic regression.

[1]: J. Mairal and B. Yu, “Complexity analysis of the lasso regularization path,”
    in Proceedings of the 29th International Conference on Machine Learning,
    Edinburgh, United Kingdom, Jun. 2012, pp. 1835–1842. [Online]. Available:
    https://icml.cc/2012/papers/202.pdf

---

> ### Author Response · Authors · 2022-08-02
> **Author response to review**
>
> We thank the reviewer for their time spent reviewing our work.
>
> We clarify the major misconception that our work overlaps with [1] - as discussed below, both the problems considered and set of techniques proposed are completely distinct. The homotopy methods for lasso in [1,2,3,4] are not directly relevant to our work since our main contribution is to generalization and online regret guarantees for unseen datasets, while [1,2,3,4] deal with computational aspects for a given seen dataset (we already include the seminal LARS paper for this orthogonal aspect).
>
> We hope the reviewer will take this into account and update their review.
>
> &nbsp;
> **Main review**
> 1. [*The literature section on exact (homotopy) methods for the elastic net and lasso is lacking and only insofar as I can see included the LARS paper at the moment. … Consider amending the literature section in the paper by including some or all of [1, 2, 3], and [4] regarding homotopy methods for the lasso.*]
> The main focus of the paper is sample complexity and online regret guarantees, exact computational aspects are orthogonal to the contributions here. We thank the reviewer for pointing out the literature on computational perspective, and we are happy to include it for the interested reader.
> 2. [*Lemma 2.1 seems very similar to the work in [1] and it is not quite clear what the novelty and enhancements of the current work are… In particular there seems to be overlap with [1] that isn't accounted for in the work.*]
> [1] i.e. [Mairal and Yu, 2012] provides refined upper and lower bounds on the number of pieces in the LASSO solution paths (e.g. compared to the upper bounds of [Tibshirani 2013] that we have used). The piecewise linear variation of beta^* with 1D parameter lambda_1 is well-known (not specific to [1]). On the other hand we provide a novel structural result for variation of EN (elastic net) loss with 2D parameters (lambda_1 and lambda_2) as a piecewise rational function with polynomial boundaries, which is completely new and useful in obtaining novel generalization and online regret guarantees for the problem. We do not see any overlap of our results with [1] – common fundamental results (e.g. [Fuchs 2005]) are used to obtain completely different results.
> 3. [*The bound on the number of problems needed for a certain accuracy in estimating λ1 and λ2 would be a useful addition, although it is not clear that the bound is useful in practice.*]
> Theorem 3.2 provides such a bound (epsilon denotes accuracy). We discuss applications to practically used scenarios of cross-validation in lines 275-288.
> 4. [*Section 4 does not seem to address a useful problem. Classification with the ordinary (least-squares) lasso is not a standard (or indeed useful) approach to classification.*]
> We respectfully disagree with the usefulness comment. Sparsity and generalization may be desirable in classification as well, e.g. [RNCR15].
>
> &nbsp;
> **Questions:**
> 1. [*The authors in [1] state that the regularization path has at worst (3^p+1)/2 segments. But you say, if I am understanding correctly, that it is exactly 3^p (l. 587). Perhaps I am missing something, but why is your result different? Should there be an O there?*]
> l.587 states “at most” and not “exactly” 3^p segments which is consistent since [1] only gives a slightly tighter upper bound. The difference in constant factor (of 2) in the upper bound is not important for our analysis.
> 2. [*Why is there only a proof sketch and not a full proof for theorem 3.1?*]
> The proof sketch in the main body is to improve readability. The full proof is included in the appendix C.1. We will mention this close to Theorem 3.1 (we already do this in lines 243-244).
> 3. [*If I understand correctly, you arrive at your bound on the number of problems needed by assuming the worst-case situation in terms of path complexity. What is the impact of this on the tightness of the bound (and practical usefulness)? I would assume that this would make the bound conservative (since in practice no paths for real data contain 3^p segments).*]
> Our results give a (polynomial) upper bound on the number of samples needed for learning the EN parameters, which holds even for worst-case problems. As the reviewer notes, fewer samples may suffice in practice.
>
> &nbsp;
> **Limitations:**
> 1. [*Include at the very least [5] in your literature section on hyper-parameter tuning for the lasso.*]
> We thank the reviewer for pointing us to [5] which provides a gradient-descent based heuristic for tuning the lasso without any theoretical guarantees. We are happy to include this along with other empirical/heuristic works that we cite, note that we do include careful comparison of our results with prior work with theoretical guarantees.
>
> &nbsp;
> **References:**
> [RNCR15] Rao, Nikhil, Nowak, Cox, Rogers. "Classification with the sparse group lasso." IEEE Transactions on Signal Processing 64, no. 2 (2015): 448-463.

---

> > ### Comment · Reviewer_zfht · 2022-08-03
> > **Response to rebuttal**
> >
> > > We clarify the major misconception that our work overlaps with [1] - as discussed below, both the problems considered and set of techniques proposed are completely distinct. The homotopy methods for lasso in [1,2,3,4] are not directly relevant to our work since our main contribution is to generalization and online regret guarantees for unseen datasets, while [1,2,3,4] deal with computational aspects for a given seen dataset (we already include the seminal LARS paper for this orthogonal aspect).
> >
> > My argument is that the computational complexity of the path is directly related to your work. I may be mistaken in this regard, but I am still not completely swayed by your argument. As I see it, section 2.1 is key to your paper and seems to be directly related to the complexity of the elastic net path homotopy solution.
> >
> > I do, however, agree that my initial assessment was too critical and will revise it accordingly.
> >
> > > On the other hand we provide a novel structural result for variation of EN (elastic net) loss with 2D parameters (lambda_1 and lambda_2) as a piecewise rational function with polynomial boundaries, which is completely new and useful in obtaining novel generalization and online regret guarantees for the problem.
> >
> > It is trivial to recast an elastic net problem as a lasso problem, so it is doubtful to me that extending theory from the lasso to the elastic net represents something that is "completely new".
> >
> > > Theorem 3.2 provides such a bound (epsilon denotes accuracy). We discuss applications to practically used scenarios of cross-validation in lines 275-288.
> >
> > You're right that you do talk about application here, but, as I wrote in my initial review, I would like to see is a more direct application of your work. Specifically, I would like to see your theorem applied to a real data set, for instance for cross-validation. If your results answer the question "how much cross-validation is enough", then you should be able to show this empirically. My concern is that your bound is not useful in practice because it is too conservative and so far it does not seem to me that you have attempted to convince me otherwise.
> >
> > > We respectfully disagree with the usefulness comment. Sparsity and generalization may be desirable in classification as well, e.g. [RNCR15].
> >
> > You misunderstand me. There is no debate as to whether sparsity is important in classification. I very much agree that it is. What I am saying is that classification by the ordinary (least-squares) lasso is not a standard or useful model. If you are able to argue that classification via $\ell_1$-regularized least-squares regression is ever preferable to $\ell_1$-regularized logistic regression (other than for computational aspects) I will rest my case.

---

> > > ### Author Response · Authors · 2022-08-05
> > > **Follow-up discussion**
> > >
> > > Thank you for responding to our rebuttal and for explaining your concerns in more detail. We believe the main source of confusion is that while it is sometimes indeed easy to extend Lasso results to ElasticNet by encoding the Ridge penalty in the dataset, since we are tuning this penalty we have to contend with the encoding changing as we change \lambda_2. This makes analyzing the problem when \lambda_1 and \lambda_2 vary simultaneously highly nontrivial. In particular, past bounds on the complexity of the solution path for fixed \lambda_2 are not enough as we have infinitely many solution paths as we vary \lambda_2.  We will revise the draft to clarify this and go into further detail below.
> > >
> > > &nbsp;
> > > [*My argument is that the computational complexity of the path is directly related to your work. I may be mistaken in this regard, but I am still not completely swayed by your argument. As I see it, section 2.1 is key to your paper and seems to be directly related to the complexity of the elastic net path homotopy solution.*]
> > > 1. We are more concerned with a specific structural complexity of the objective (specifically that the dual loss function is piecewise structured in the sense of Definition 1) which has implications for the learnability of the hyperparameter across instances. In particular, generalizability only depends on this structure, and optimization techniques to compute the optimal hyperparameters constitute an orthogonal direction in our view. While this structure uses path complexity results for Lasso, it is particularly non-trivial for the ElasticNet (see our response to the next question).
> > > 2. In more detail, we need to bound the number of sign patterns on N problem instances (in the sense of Definition 4) in order to bound the number of problems for which all sign-patterns are possible (i.e. are shattered). Such bounds for ElasticNet do not follow from bounds on the Lasso solution path complexity (or ‘ElasticNet solution path’ for fixed lambda_2) since different values of lambda_2 can result in uncountably many different solution paths.
> > > 3. For our online learning results, we need further properties beyond the piecewise structure (non-trivial even for the special case of Lasso). We need to show that transition boundaries of the piecewise structured dual function do not concentrate in a small region of the parameter space in the sense of Definition 3 as otherwise online learning is not possible (Balcan et al., 2018).
> > >
> > > &nbsp;
> > > [*It is trivial to recast an elastic net problem as a lasso problem, so it is doubtful to me that extending theory from the lasso to the elastic net represents something that is "completely new".*]
> > > 1. The main novelty is to characterize the structural complexity of the loss when lambda_2 is also simultaneously varied.  While the standard reduction creates a different Lasso problem for each fixed value of lambda_2, our structural results answer how the infinitely many ElasticNet solution paths are related as lambda_2 is continuously varied. We believe our structural results are new and nontrivial given prior work.
> > > 2. Prior computational results and the reduction imply efficient computability of solutions for fixed lambda_2; we provide a structural result (without addressing computability) for continuous values of lambda_2. This extension is essential for obtaining our sample complexity and online learning results.
> > >
> > > &nbsp;
> > > [*I would like to see is a more direct application of your work. [...] My concern is that your bound is not useful in practice because it is too conservative and so far it does not seem to me that you have attempted to convince me otherwise.*]
> > > We only claim to provide provable upper bounds on the sample complexity for worst-case instances, i.e. provide sufficient (but not necessary) conditions for learnability. We believe our theoretical results are important in their own right, and implementation comes with computational challenges, resolving which is an interesting future direction.
> > >
> > > &nbsp;
> > > [*[C]lassification by the ordinary (least-squares) lasso is not a standard or useful model. If you are able to argue that classification via ℓ1-regularized least-squares regression is ever preferable to ℓ1-regularized logistic regression (other than for computational aspects) I will rest my case.*]
> > > We were interested in the setting in Section 4 because the ordinary Ridge regression subproblem *is* used for classification in-practice: it is [implemented in scikit-learn](https://scikit-learn.org/stable/modules/generated/sklearn.linear_model.RidgeClassifier.html) and has been the learning algorithm of choice in well-known prior work (Rahimi & Recht, 2007). This led us to work on the more general ElasticNet problem.
> > >
> > > &nbsp;
> > > **References:**
> > > 1. Balcan, Dick, Vitercik. *Dispersion for data-driven algorithm design, online learning, and private optimization*. FOCS 2018.
> > > 2. Rahimi, Recht. *Random features for large-scale kernel machines*. NeurIPS 2007.

---

> > > > ### Comment · Reviewer_zfht · 2022-08-08
> > > > **Another Response**
> > > >
> > > > Thank you for the additional comments and clarification. I think I see your point and have updated my review again. I still contend that classification via the ordinary elastic net is not relevant (simply being implemented in a software suite and having been used in a well-known reference is not a strong argument), but it is not a major point. I also feel that the issue of applicability is not yet addressed, however, and I urge you to consider adding a practical example to the final revision.

---

### Official Review · Reviewer_yYWW · 2022-07-10

**Rating:** 7
**Confidence:** 3
**Soundness:** 3 good
**Presentation:** 3 good
**Contribution:** 3 good

**Summary:**

This paper studies the validation loss of the elastic net as a function of its two hyperparameters, $\lambda_1, \lambda_2$. The authors allow "validation loss" to be fairly general: it is a function that maps a dataset and a setting of $(\lambda_1, \lambda_2)$ to the validation loss of the elastic net. They characterize this function as a piecewise rational polynomial as a function of $(\lambda_1, \lambda_2)$. They use this characterization to show that, given a set of validation losses and a distribution over this set (e.g. in leave-one-out cross-validation, the validation losses are the leave-one-out losses with the uniform distribution), a possibly small number of draws from this distribution are needed to minimize the average validation loss within $\epsilon$ accuracy. The authors go on to show that validation losses can be minimized in an online setting, where we view each validation loss one-by-one. Finally, the authors prove similar results for binary classification, where classification is done by thresholding the predictions of the elastic net.

**Questions:**

- Can the authors provide a proof for the statement "using simple algebra, the denominator..."?

I also have a few questions relating to the meaning of the author's results.
- Is the algorithm in Theorem 3.3 just Algorithm 1? It's probably worth stating this if so. As-is, Algorithm 1 isn't really discussed in the paper.
- One of the claims of Theorem 3.3 is about the $\beta$-dispersion of a series of functions. Why is this something readers should care about (i.e. why have the authors put this in the theorem's statement)?
- On lines 235-237, the authors state that they have solved the problem of optimizing the complex validation loss as a function of $(\lambda_1, \lambda_2)$. Which of their results back up this statement? It seems like Theorem 3.2 is about the number of validation samples required, and Theorem 3.3 has error growing with the square root of the number of samples.
- The authors state that their "assumption of smoothness [on $y$] is much weaker than sub-Gaussian noise assumptions in the literature." But this smoothness assumption is paired with assuming $y$ is a bounded random variable; isn't this much weaker than sub-Gaussianity, as boundedness implies sub-Gaussianity?
- What exactly does it mean in Theorem 3.3 that $l_1, \dots, l_T$ are "independent"? Are we assuming the existence of some distribution $\mathcal{D}$ over losses, and $l_t$ are sampled i.i.d. from $\mathcal{D}$? This should be stated more precisely.

**Limitations:**

I think the authors have addressed the limitations of their work, with the exception of a few of the points already discussed above.

**Strengths And Weaknesses:**

**Strengths**
I think the technical contributions of this paper are pretty strong and original. As far as I know, no one has given this exact a characterization of the loss landscape of things similar to cross-validation for the elastic net. I can see some interesting future work building off of this paper -- e.g. what do the individual pieces of the loss look like? The author's observation about how much subsampling of leave-one-out CV is needed to get an accurate approximation is interesting as well. I actually think this is one of the most immediately practical pieces of advice from this paper, and might warrant a little more highlighting (e.g. as a corollary). I can't really comment on the strengths of their results about online learning, as I'm not as familiar with that literature (in particular, $\sqrt{T}$ expected regret for $T$ problems sounds like a lot, but maybe this is near the best you can do?).

**Weaknesses**
I think there are two main directions for improvement of the paper. First, I think the authors have slightly oversold the weakness of their assumptions, and should tone down those claims a little bit. Second, the writing was pretty dense and could use some more discussion and careful definitions.

On the assumptions required for the authors' results:
- The abstract states that the results do not require "strong assumptions on the data distribution". Assumption 1 (that the covariates and responses are all bounded) is definitely not a weak assumption. I think it wouldn't be unfair to say this is a "strong" assumption, as this disallows, say, a well-specified model in which responses are linear functions of the covariates with Gaussian noise.
- The results are stated as being completely independent of data distribution (e.g. in "Theorem 1.1 (Informal)"). However, a number of the lemmas used (Lemmas B.2, B.3, and C.1) require the columns of the data matrix $X$ to be in general position. First, this condition should be stated in the results using these lemmas (e.g. in the statement of Lemma 2.1). Second, it should be clarified in exposition that the paper's results only apply to sets of problems / distributions over problems with covariate matrices with columns in general position.


On the writing being dense: The paper is pretty technical and uses a number of concepts from algebraic geometry and learning theory. These concepts are sometimes unexplained (e.g. just a definition is given) or lack a definition altogether. I think the results in this paper could be of interest beyond the learning theory community or algebraic statistics community, and so these concepts should be better explained (I think this is especially true of the concepts from algebraic geometry, which I suspect very few NeurIPS readers / attendees have experience with). Here are the things that stood out to me:
- Piecewise structured functions could use a simple English description.
- Pseudo-dimension should be defined and briefly explained. The abbreviation PDIM should also be explicitly defined.
- Semi-algebraic sets and algebraic curves should be defined.
- A precise statement of Bezout's theorem should be given in the form that is applied here (or, at least, a reference).
- I thought the definition and explanation of $\beta$-dispersion was really clear and helpful.

**Miscellaneous things**
- I think Lemma C.2 needs some conditions to ensure $A^T A + \lambda I$ is invertible (what if $A$ is low rank? what if $\lambda = -5345$?)
- Line 215: "based on the observation above" there are many observations preceding this statement; it's not immediately clear which one this refers to.
- What are the asterisks in the proof of Lemma 2.1 (around line 218)? Are these just typos? An asterisk appears on $\lambda_1$ around line 223 as well, and then disappears in the next equation.
- "Using simple algebra, the denominator..." Definitely $\mathcal{E}$ depends on $\lambda_1$ and $\lambda_2$, and it's not immediately clear to me why that dependence gets canceled out in this expression. I think a proof should be given here.
- Algorithm 1 is actually never discussed in the paper. I'm guessing it's the algorithm from Theorem 3.3, but this should really be made clear (and discussed).

---

> ### Author Response · Authors · 2022-08-02
> **Author response to review**
>
> We thank the reviewer for their time spent reviewing our work.
>
> **Main review**
> 1. [*$\sqrt{T}$ expected regret for T problems sounds like a lot, but maybe this is near the best you can do?*]
> Indeed $\sqrt{T}$ is the best one can attain (see e.g. Cesa-Bianchi and Lugosi, Theorem 3.7).
> 2. *Weakness of assumptions*:
> We remark that the boundedness assumption (Assumption 1) is only needed for the online learning results.
> 3. *Writing*:
> We thank the reviewer for suggestions on adding the relevant definitions from learning theory and algebraic geometry to improve readability for the general reader. We will make sure these suggestions are incorporated.
>
> &nbsp;
> **Questions:**
> 1. [*I think Lemma C.2 needs some conditions to ensure ATA+λI is invertible (what if A is low rank? what if λ=−5345?)*]
> Good point, assuming the parameter \lamdba to be positive is a sufficient condition for invertibility since A^TA is positive semi-definite. We will state this explicitly.
> 2. [*Line 215: "based on the observation above" there are many observations preceding this statement; it's not immediately clear which one this refers to.*]
> This is the immediately previous observation in Lines 209-212, we will make this more clear.
> 3. [*What are the asterisks in the proof of Lemma 2.1 (around line 218)? Are these just typos? An asterisk appears on λ1 around line 223 as well, and then disappears in the next equation…. Can the authors provide a proof for the statement "using simple algebra, the denominator..."?*]
> The asterisk terms are the same as those defined in (proof of) Lemma C.1 which is invoked right before line 218, we apologize for the omission here. There is indeed a typo in the denominator expression which we have fixed now and elaborated on the steps for the algebraic simplification for additional clarity.
> 4. [*Is the algorithm in Theorem 3.3 just Algorithm 1? It's probably worth stating this if so. As-is, Algorithm 1 isn't really discussed in the paper.*]
> That is correct. We briefly discuss Algorithm 1 in lines 327-329 right before Theorem 3.3, but will also explicitly state it in the theorem statement. Thanks for the suggestion.
>
> 5. [*One of the claims of Theorem 3.3 is about the β-dispersion of a series of functions. Why is this something readers should care about (i.e. why have the authors put this in the theorem's statement)?*]
> We note in lines 296-297 that prior work shows that dispersion is necessary and sufficient for online learning of piecewise Lipschitz functions. We will also re-emphasize this in context of Theorem 3.3 for clarity.
> 6. [*On lines 235-237, the authors state that they have solved the problem of optimizing the complex validation loss as a function of (λ1,λ2). Which of their results back up this statement? It seems like Theorem 3.2 is about the number of validation samples required, and Theorem 3.3 has error growing with the square root of the number of samples.*]
> This refers to Theorem 3.3. Note that the expected regret in the online setting, when averaged over the $T$ samples, is $1/\sqrt{T}$ which is at most $e$ for $T\ge 1/e^2$.
> 7. [*The authors state that their "assumption of smoothness [on y] is much weaker than sub-Gaussian noise assumptions in the literature." But this smoothness assumption is paired with assuming y is a bounded random variable; isn't this much weaker than sub-Gaussianity, as boundedness implies sub-Gaussianity?*]
> This is a great question. We first remark that the assumptions are not needed at all for our distributional results. Coming to the present question, indeed boundedness implies sub-Gaussianity but only by setting sigma to be the range of the bounded variable which can be a very loose estimate of the bounded variable. On the other hand, $\kappa$-bounded smoothness includes a much larger class of distributions (since we do not need the strong tail decay of sub-Gaussianity) which includes (truncated) sub-Gaussian distributions in particular. We agree with the reviewer that our current statement needs clarification, we propose to include the above discussion.
> 8. [*What exactly does it mean in Theorem 3.3 that l1,…,lT are "independent"? Are we assuming the existence of some distribution D over losses, and lt are sampled i.i.d. from D? This should be stated more precisely.*]
> The losses are independent but not identically distributed in Theorem 3.3. That is, the losses are random functions and the random coin flips for generating the losses are independent. We will add this clarification, thanks for asking this.

---

### Official Review · Reviewer_MWke · 2022-07-13

**Rating:** 8
**Confidence:** 4
**Soundness:** 4 excellent
**Presentation:** 4 excellent
**Contribution:** 4 excellent

**Summary:**

This submission derives new generalization bounds for tuning the regularization parameters for ridge-regression, LASSO, and the elastic net methods.
In particular, the authors study two settings: (i) tuning based on multiple draws of train/test splits from a fixed (but arbitrary) problem distribution and (ii) online learning, where new train/test splits are provided as problem instances at each time step.
In the former setting, the authors show $\tilde O(p^3/\\epsilon^2)$ problem samples are required to come within $\\epsilon$ of the best (expected) loss.
In the latter setting, the authors impose a smoothness assumptions on the adversary in order to provide a continuous version of the multiplicative weights algorithm with sub-linear regret.
Both results rely on a novel characterization of the tuning objective as a piecewise rational function with boundaries given by polynomial curves.
The submission concludes with extensions to linear classification problems with a fixed decision threshold and squared loss.

**Questions:**

- Theorem 3.2/Theorem A.1: Is this result constructive or does it only give the existence of an algorithm?
    For instance, I agree with the claim that "our results answer the question ‘how much 288 cross-validation is enough’ to effectively implement the above techniques", but this is separate from carrying out the optimization procedure in practice.

- Algorithm 1: The optimization domain $C$ appears to be an uncountable set, in which case the density function $w_i$ defines an improper distribution (at least initially) and maybe highly discontinuous during execution of the algorithm.
    As a result, Algorithm 1 appears to be unimplementable --- is this correct?

- Are there any lower bounds for the sample complexity of learning $\lambda$ in the distributional setting (regression of classification)? If so, it would be excellent to compare Theorem 3.2 to these rates. If not, it might be nice to comment on the challenges in developing these bounds.

**Limitations:**

The limitations are properly addressed.

**Strengths And Weaknesses:**

This is a highly novel paper which tackles an excellent problem: how to properly tune regularization parameters for common algorithms.
The standard approach to tuning ridge regression, LASSO, or the elastic net methods is grid-search.
However, as the authors note in the classification setting, some model choices, like thresholding, can induce discontinuities in the tuning objective and make grid-search unreliable.
Thus, developing better alternatives to grid-search is a important topic.

The theoretical bounds are novel and cover a range of important settings.
I was particularly interested to see the online-learning results.
The only two potential issues I see with this work are implementability and the focus on problem-sample complexity, rather than example-sample complexity.
The apparent non-implementability of the actual methods for tuning the regularization parameters is somewhat disappointing, but not too surprising given the nature of the results.
And, while the authors state that problem-sample complexity is more common, I think the problem sampling (which covers cross-validation) to be far more interesting.

### Writing

The paper is very well written.
I found only a few typos (see Minor Issues below).
I particularly appreciate that the authors take care to give intuition for each theoretical result as well as discuss their implications.
Congratulations on a very well-written paper.

### Theory

**Soundness**: This paper is outside of my core research area, so I'm not an expert on the theory. However, I briefly looked through the appendices and saw no major issues.

**Implementability**: My understanding from the discussion is that the methods for tuning $\\lambda$ are not implementable (see Questions).
Thus, while the paper presents "efficient algorithms" (Line 402), efficiency is only in the sense of sample complexity, rather than computational complexity.
I don't think this is an important issue, but I suggest the authors either comment on the complexity of tuning $\\lambda$ in practice, or clarify that they mean efficiency only in terms of samples.


### Experiments

N/A


### Minor Comments and Typos:

- Line 263: "rational functions of bounded degrees ." -- remove extra space before the period.
- Line 286: "$\epsilon$-approximation of the loss corresponding parameter selection with arbitrarily..."  --- I don't understand this sentence; perhaps a word is missing?
- Line 369: "only polynomially many problem samples to generalize well..." -> "only polynomially many problem samples are required to generalize well"

---

> ### Author Response · Authors · 2022-08-02
> **Author response to review**
>
> We thank the reviewer for their time spent reviewing our work.
>
> 1. [*I suggest the authors either comment on the complexity of tuning λ in practice, or clarify that they mean efficiency only in terms of samples.*]
> Thank you for the suggestion. We will add a small discussion to clarify this. We will also address the typos pointed out by the reviewer.
> 2. [*Theorem 3.2/Theorem A.1: Is this result constructive or does it only give the existence of an algorithm? For instance, I agree with the claim that "our results answer the question ‘how much cross-validation is enough’ to effectively implement the above techniques", but this is separate from carrying out the optimization procedure in practice.*]
> The result in Theorem 3.2 gives polynomial sample complexity (in p, epsilon, delta) for learning lambda using the Empirical Risk Minimization (ERM) algorithm for which a polynomial time implementation is not specified. This (using loss minimizer on the sample) is typical when giving generalization guarantees by bounding pseudo-dimension or VC dimension, we are happy to add this clarification.
> 3. [*Algorithm 1: The optimization domain C appears to be an uncountable set, in which case the density function wi defines an improper distribution (at least initially) and maybe highly discontinuous during execution of the algorithm. As a result, Algorithm 1 appears to be unimplementable --- is this correct?*]
> Good question. Yes, there may be a large number of discontinuities in the worst case, so a polynomial time implementation (if possible) of Algorithm 1 is not known.
> 4. [*Are there any lower bounds for the sample complexity of learning λ in the distributional setting (regression of classification)?*]
> We believe this is a very interesting question which we haven’t explored in the present work, and lower bounds are not known for our setting to the best of our knowledge.

---

> > ### Comment · Reviewer_MWke · 2022-08-05
> > **Author Response**
> >
> > Thanks for addressing my comments and answering my questions.
> >
> > - I'm glad we agree about lower bounds; it would be very interesting to know the limits of estimation in this setting.
> >
> > - It would be nice cool to see implementable algorithms in this setting, although the current results are an excellent start.

---

### Meta-Review · Area_Chair_d7fJ · 2022-08-24

**Recommendation:** Accept
**Confidence:** Certain

**Metareview:**

The reviewers agreed that this paper should be accepted -- it studies an interesting and in someways "overlooked" problem and seems like it could be a starting point for others to build on. The paper did have some weaknesses. For example, we feel that the lack of experimental results is a missed opportunity -- the reviewers feel the paper would be made stronger by including at least a simple example where the standard approach of grid search + CV fails. Without such an example, it is a bit hard to be convinced of the importance of the problem being studied.

**Award:**

No

---

### Decision · Program_Chairs · 2022-09-14

Accept